# “It’s Not That We Care Less”: Insights into Health Care Utilization for Comorbid Diabetes and Depression among Latinos

**DOI:** 10.3390/ijerph21020148

**Published:** 2024-01-29

**Authors:** Sharon Borja, Miriam G. Valdovinos, Kenia M. Rivera, Natalia Giraldo-Santiago, Robin E. Gearing, Luis R. Torres

**Affiliations:** 1Graduate College of Social Work, University of Houston, Houston, TX 77004, USA; rgearing@central.uh.edu; 2Graduate School of Social Work, University of Denver, Denver, CO 80208, USA; miriam.valdovinos@du.edu; 3Department of Psychology, University of Denver, Denver, CO 80208, USA; kenia.rivera@du.edu; 4Department of Psychiatry, Massachusetts General Hospital, Boston, MA 02114, USA ngiraldosantiago@mgh.harvard.edu; 5School of Social Work, University of Texas, Rio Grande Valley, Edinburg, TX 78539, USA; luis.torreshostos@utrgv.edu

**Keywords:** diabetes, depression, comorbidity, Latinos, healthcare utilization, United States

## Abstract

Despite robust knowledge regarding the socio-economic and cultural factors affecting Latino* access to healthcare, limited research has explored service utilization in the context of comorbid conditions like diabetes and depression. This qualitative study, embedded in a larger mixed-methods project, aimed to investigate perceptions held by Latinos and their social support systems (i.e., family members) regarding comorbid diabetes and depression and to identify barriers and facilitators to their help-seeking behaviors and treatment engagement. Bilingual and bicultural researchers conducted eight focus groups with 94 participants in a large U.S. metropolitan area and were primarily conducted in Spanish. The participants either had a diagnosis of diabetes and depression or were closely associated with someone who did. This study identified key individual and structural barriers and facilitators affecting healthcare access and treatment for Latinos living with comorbid diagnoses. A thematic analysis revealed structural barriers to healthcare access, including financial burdens and navigating healthcare institutions. Personal barriers included fears, personal responsibility, and negative family dynamics. Facilitators included accessible information, family support, and spirituality. These findings underscore the need to address these multi-level factors and for healthcare institutions and providers to actively involve Hispanic community members in developing services and interventions.

## 1. Introduction

Diabetes is a significant public health problem, affecting 11.3% of the United States (U.S.) population [1] and accounting for over eight million hospitalizations annually [2]. Roughly 20% of previously hospitalized patients with diabetes were re-admitted within 30 days, 26% were re-admitted within three months, and 30% were re-admitted within a year [2]. Racial and ethnic minorities are at a higher risk of developing diabetes, with Latinos (for the purposes of this paper, we use “Latino” to refer to individuals from North, South, and Central America or with this cultural heritage, including those who identify as ”Latinx,” ”Chicana/o,” ”Tejana/o, ” or ”Hispanic”) accounting for nearly 12% of all diabetes cases in the U.S. and being twice as likely as non-Hispanic Whites to develop type 2 diabetes [3]. Furthermore, Latinos are three times more likely to have diabetes-related end-stage renal disease [4] and 1.2 times more likely to undergo amputation than non-Hispanic Whites [5]. Moreover, Latina women are 1.4 times more likely to die from diabetes compared to their non-Hispanic White counterparts [6].

Mental health comorbidities (i.e., depression co-occurring with diabetes) are also a growing concern. Poor glycemic control, especially with the severe and rapid progression of diabetes complications, has been linked to depression [7,8]. Depression is a significant risk factor for diabetes [9,10], particularly among Latino adults (8.2%) experiencing depression in the United States [11]. The co-occurrence of depression and diabetes is often underdiagnosed and undertreated as many Latinos who experience these diagnoses do not actively seek professional help because of stigma [12]. Despite the growing rates of diabetes and depression among racial and ethnic minorities, little is known about the lived experiences of Latinos with diabetes co-occurring with depression, their families’ perspectives, and how these diagnoses and relationships shape their engagement in healthcare services.

### Barriers to and Facilitators of Healthcare Access

Access to healthcare services is critical to achieving and maintaining good health. We define access to care as the ability to receive effective services from a health provider within a satisfactory timeframe and to continue engaging in treatment as needed. This definition encompasses two crucial dimensions of healthcare access: having (or not having) access to and utilizing (or not utilizing) treatment and services. The U.S. does not have universal healthcare [13]. As such, healthcare access is determined by a complex arrangement of public and private schemes such as Medicare for those 65 and older and some individuals with disabilities; Medicaid, which provides for those below a certain income threshold; and employer-sponsored plans for those in the workforce and their eligible dependents [13].

The 2010 Affordable Care Act brought substantial reform to the healthcare system, expanding Medicaid in some states and establishing a health insurance marketplace with government-subsidized health plans [14]. Despite these reforms, insurance access remains uneven, and obtaining health insurance remains a significant structural barrier to healthcare access and engagement in the Latino community. For example, uninsurance rates are markedly higher for Latino adults (17.7%) and non-Hispanic Blacks (9.6%) than for non-Hispanic Whites (5.7%) [15]. Latinos also continue to be overrepresented in low-service-utilization groups and the premature termination of services [16]. Other established structural barriers to mental health concerns within Latino communities include limited childcare, transportation, English-only services, and the cost of services [17,18]. These barriers often interact with racial disparities in access to mental health services, exacerbating health inequity [11,19]. The structural roots of these inequalities must be understood to address critical gaps in service access and utilization among Latinos.

Individual, social, and cultural factors also impede Latinos from receiving appropriate and timely health services [12]. Language, stigma, types of symptoms or disorders, cultural beliefs, acculturative stress, length of time in the US, and poverty have been identified as reasons Latinos are less likely to reach out and engage with health providers [11,17,20,21]. Evidence from qualitative studies indicates that mental health needs (such as depression) could also be implicated in the reduced participation of Latinos in healthcare services. Among immigrant Latinos with type 2 diabetes, emotional suffering (e.g., sadness) has been found to impede treatment engagement [22]. Despite increasing awareness regarding factors shaping Latino access to care, very few studies have considered how these factors interact in the context of comorbid conditions, especially co-occurring diabetes and depression.

This study explored the experiences of Latinos with diabetes and comorbid depression and their families, focusing on two dimensions of access to care: access and utilization. As such, help-seeking and treatment engagement were central to the two-fold objective of this study. First, we sought to explore how Latino patients and their social support systems navigate living with comorbid conditions of diabetes and depression. Our second objective was to examine the barriers and facilitators of help-seeking and treatment engagement and to understand the role of social support systems. Through these objectives, we attempt to contextualize Latino experiences of healthcare access within comorbid conditions and explore the factors contributing to their access and sustained treatment engagement. Considering the perspectives of both family members and individuals living with comorbid conditions will be crucial for developing treatments that fully meet the social and cultural needs of this population.

## 2. Materials and Methods

This qualitative study was embedded in a larger mixed-methods research project that sought to understand the experiences, attitudes, and perceptions of Latinos and their social support toward comorbid diabetes and depression. This article presents the qualitative findings from eight focus groups conducted with Latinos residing in a large metropolitan area in the U.S. who have experienced these issues or have family members who have had these experiences.

### 2.1. Sample

We recruited participants from public settings (e.g., community agencies, local churches, etc.) in a large metropolitan area in the South of the U.S. Recruitment materials were disseminated in Spanish and English through agency staff, church ministers, and elders who distributed the materials to clients and members. The inclusion criteria required that all participants self-reported diagnoses of diabetes and depression or were closely associated with someone who does. Ninety-four individuals participated in eight focus groups, of whom 67 (71%) self-identified as women and 27 (29%) as men. The participants represented several Latin American countries, including Colombia, Cuba, El Salvador, Guatemala, Honduras, and Mexico. A convenience sample was used with individuals who attended the focus group sessions.

### 2.2. Data Collection

We invited 12 participants for each focus group, anticipating that not everyone would attend. Unexpectedly, everyone showed up, and some participants also brought additional guests. Focus group sessions were facilitated by bilingual and bicultural facilitators, including the principal investigator, a co-investigator, and two graduate research assistants. Standard recommendations for the number of focus group participants range from five to ten, but six to eight are preferred [23]. We welcomed everyone who met the inclusion criteria, making the focus larger than initially planned. Given the larger-than-expected size of the groups, the facilitators ensured that each participant had the opportunity to share in the discussions.

The focus groups were conducted at the facilities of our primary community partners. Seven sessions were conducted in Spanish and one in English. The focus group discussions allowed participants to share similar and unique experiences. Previous studies on self-disclosure suggest that participants are more willing to share about themselves with those who share similar characteristics than with those who differ from them [23]. The focus groups were recorded on audio tape after signed consent was obtained. Demographic information was anonymously collected for each participant through group tallies. An independent contractor transcribed the recordings, after which the audio files were stored on a password-protected computer in the principal investigator’s office and were only accessible to the research team. 

The facilitators used a semi-structured focus group guide developed by the research team to elicit participants’ stories about their experiences with comorbid diabetes and depression and the factors that affect their ability to access care and engage in treatment. It consisted of nine questions: two for an introduction, two were transitional, three were main discussion points, and two to obtain final comments and closure. This number of questions aligns with the recommended six to ten questions for focus groups lasting from 90 min to 2 h [23].

### 2.3. Data Analysis

The transcribed recordings were translated by two graduate assistants who were bilingual (Spanish and English) and bicultural. One of the co-investigators reviewed the translations for accuracy and back translation. Then, we conducted thematic analyses to analyze the transcriptions and field notes of the focus group moderators. Open coding was used to identify codes and themes. An open coding technique was used to develop a codebook of descriptive codes coinciding with the focus group guide categories. Key categories were applied to the transcript sections that addressed the research questions (e.g., access barriers and treatment facilitators). The research team generated a codebook through an iterative coding process. At least two coders reviewed each focus group transcript line by line, and the research team mutually agreed upon the final codes in weekly analysis meetings. Categories were refined, and all coders agreed upon sub-themes emerging from the analysis. The process was iterative, with continuous reviews and revisions of codes and themes while using the online analytic software Dedoose, Los Angeles, CA, USA.

## 3. Results

The participants’ descriptions of their experiences with health problems demonstrated the chronicity and severity of their challenges. Stories of severe physical complications related to diabetes, such as amputations and death, highlight the magnitude of these issues for participants. A discussion about the high prevalence of diabetes in the Hispanic community triggered a conversation on whether diabetes was an inherited health condition and its intergenerational implications, specifically for children.

Themes of a structural and personal nature related to barriers to healthcare access and facilitators of sustained engagement in treatment emerged. The sub-themes that shaped the health-related decision-making of participants are summarized in Table 1 and further elaborated in this section.

### 3.1. Structural Barriers

#### 3.1.1. Financial Burden

Participants identified financial burden as a structural barrier to healthcare access and engagement, especially concerning diabetes and comorbid mental health problems. Personal accounts of economic difficulties were also discussed. However, participants also recognized the macroprocesses that implicate the U.S. healthcare system as a medical–industrial complex imposing financial burdens on those with chronic health conditions. One participant said,

*“Here in the United States, medicine is a business, and the business is for those who have money, not for those who do not have it. People do not go and won’t go to see a doctor, and all this will continue as long as there are not enough funds to help Hispanics and other poor people.”* (Member from focus group 5).

Related to financial burden, a lack of health insurance was a common thread impacting access to proper medical care. Uninsured status made treatment decisions more difficult and health institutions more challenging to navigate. The following quote highlights the reality of making difficult choices given one’s limited financial resources:

*“I never went for an exam for prevention—never, never. Why? It has to do with this [makes money gestures with fingers]: if you don’t have health insurance, they will charge you for the exams. So, I have to put this into balance and say, “Do I put bread on the table for my family, or do I go to the doctor?” What do I do? I have to put food on the table.”* (Member from focus group 1)

Similarly, participants underscored how the type of employment they held imposed further constraints on their ability to take time off to consult a medical provider to address health issues. Per one participant,

*“I have lived here for 16 years. Many people do not go to the doctor. They do not have the hours at work. If they miss a day, you are told, ’You are fired’... or you cannot go because you are needed in your section; so many people cannot go to the doctor because they do not want to lose their jobs.”* (Member from focus group 4)

Furthermore, participants shared their difficulties with establishing eating habits that help reduce health risks due to the financial burden of healthier food options in their communities. One interviewee shared,

*“It’s very hard economically. People say you’ve got to diet, and then when you have a big family, it’s very hard to buy foods that are nutritional because it’s very expensive ... When you come from a big family, and you go grocery shopping, and you try to stretch your money, so you buy whatever there is that is cheaper to put food on the table. Actually, you should buy healthy food, but that is very expensive. The government doesn’t help.”* (Member from focus group 8)

#### 3.1.2. Access to Information

Another sub-theme that emerged was a lack of access to accurate information about a diagnosis, services, and treatment options. Confusion about the different types of diabetes and the use of insulin was observed in most of the focus groups. As one participant stated,

*“if you don’t know, you’re not going to be doing anything about it.”* (Member from focus group 8)

In another case, a participant was confused about why specific tests were being conducted and felt ill-prepared to ask questions about her circumstances. She shared,

*“Last time they sent me [for this procedure], but they never told me that I had to go a few days before ... I went straight to them. It wasn’t there in the clinic. It was elsewhere. ... My husband didn’t go to work to take me, but they [clinic staff] didn’t explain it to me well. They never explained to me what it was.”* (Member from focus group 7)

Often, the lack of information was complicated by the undocumented and uninsured status of the participants. One interviewee expressed frustration with this vicious cycle of not seeking help. He stated,

*“Definitely, I believe that the key is that we do not have information, and many people have not had information, but it is because of the fear of approaching to seek information, fear of approaching hospitals because there are no resources. Often, we do not have money. We don’t have insurance. We have no way of getting anywhere because we say, ’Why should I go if I don’t have insurance?’ We do not have help right away, definitely, and it is fear. We fall into the same thing, it is fear—fear for their immigration status, fear for so many things we hear [happening] around here.”* (Member from focus group 5)

#### 3.1.3. Navigating Healthcare Institutions

Another significant structural barrier was navigating complex healthcare institutions as a Latino in the U.S. The following participant expanded on her family’s societal experiences and how those experiences impacted how the family would not engage with healthcare. The participant shared,

*“All of my uncles are diabetic, and I have many diabetic cousins, but I don’t take care of myself. I eat everything, but well, it is like that here [in the US]. We are always running from one place to another. So, we cannot eat healthy because those of us who work a lot work more than eight hours. We arrive, and we make some food quick, not very nutritious... So you eat whatever you can to take the hunger away, and you don’t eat healthy.”* (Member from focus group 1)

Navigating these institutions meant experiencing discrimination and racism related to being minoritized racially and linguistically. One person shared:

*“Because we are Latino and we carry the color of our skin, they have discriminated against me. They think I don’t know anything.”* (Member from focus group 1)

In response to this person’s comment, another focus group participant said,

*“Sometimes there are people who do speak Spanish but do not want to speak it. It is the new racism that the current president [POTUS 45] is imposing. It will be worse because we are already afraid even to speak our language because you speak Spanish, and then others are turning to see you. I imagine that this means we will be less able to have help for Latinos in their language.”* (Member from focus group 7)

### 3.2. Personal Barriers

#### 3.2.1. Fear

Fear was one of the most common barriers in all focus groups. This has to do with various reasons why people are afraid to engage with or access the healthcare system. One person stated,

*“There are many barriers for us as Hispanics, mainly because of the immigration system right now. There are many people who are afraid to seek help, afraid of going to hospitals for fear of being asked about their immigration status.”* (Member from focus group 5)

Participants also described fear as directly impacting help-seeking behaviors and engagement in treatment in the early stages of the disease. Their immigration status was a common source of fear, but at other times, the apprehension to reach out and engage in health services was motivated by a fear of receiving multiple diagnoses and not having the resources to keep up with treatment. This quote offers a glimpse of some of these dynamics:

*“Many people stop going to the doctor even if they feel bad because they say the doctor will find thousands of new things in them. They start with the lab studies and analysis, and that’s where they begin to realize, ’this person is suffering from this, the other, and the other,’ and that’s where they say, ’I told you, what’s the purpose of going to the doctor? To find out that I have thousands of diseases. It’s better not to go and wait until something hurts. At that point, I will take a pill. When another thing hurts, I will take another pill.”* (Member from Focus group 5).

Another barrier was the personal responsibility participants assumed when dealing with their diagnoses. It was common to hear comments like,

*“We [Hispanics] never go to a doctor, only when we feel that we are dying.”* (Member from focus group 4)

Many focus group participants agreed with these statements, and some elaborated on them. One participant shared,

*“I think it is part of our culture to say, ‘I feel a little better now, and that’s fine’, so we stop taking the medication, stop attending consultations and doctor appointments. So, I think it’s part of not being responsible for yourself. We are harming ourselves.”* (Member from Focus group 5).

Some of these comments did not consider how the environment in which people live and the systems they interact with may influence their personal choices. However, a few participants expanded their responses and discussed how personal choices were more complicated than simply blaming the individual. The following quote offers a deeper understanding:

*“Sometimes we as Hispanics tend, as we say, ’we care less.’ They diagnose some disease, and we take it lightly. Nothing happens, or a person comes and tells us, ’Take this because this will help you.’ Sometimes, we simply do not go to the doctor, but we ignore what we really should do and do other things that we should not do, such as going around self-medicating, because that is what is common. We self-prescribe before going to the doctor. Why? Sometimes, our budget is insufficient, and sometimes, the person does not have health insurance. There are several things, but as Hispanics, we tend not to pay attention to the real stuff that we have to focus on, such as the disease, which is something essential that we have to have first in mind, that we have to take into account that we have to take care of ourselves.”* (Member from focus group 5)

#### 3.2.2. Family Dynamic

The response of family members to a participant who received a diabetes diagnosis could impede their active engagement in healthcare services and activities to improve their health status. One participant described how their partner was not supportive when trying to change their lifestyle for physical health.

*“The truth is, in my home, there has been no support from my partner. I decided to change my life because I have my children, and I have always tried to do the best for them, to work for them because I don’t want my foot severed tomorrow or they take an organ out of me because they are still little. They cannot stand on their own. On the part of my husband, I have never had his support. I have always tried, I cook this, this is what we are going to eat, but he always goes, ’No, if you give it to me, give me four portions of food,’ and not a portion like this [signals small amount], but a portion like this [signals a large amount] and I say to him, ’you are going to eat all of that?’ and he says, ’Yes, and it won’t even fill me up,’ and I tell him, No.”* (Member from focus group 2)

Participants shared feelings of shame and were criticized by their family members for needing mental health services.

*“You go to the psychologist, you’re crazy, always the negative, but therapy does help because people go for many situations. Many people may be divorcing... many are afraid to tell their family they need help because they are ashamed. Many people are ashamed that they need that help and are afraid to tell the family, ’Look, I feel like this’ because then the family is like, ’I don’t want to hear it.’ They don’t want to accept that.”* (Member from Focus group 3).

Participants also discussed how the lack of support is detrimental to their wellbeing. One participant described it briefly as

*“...every day it will get worse if there is no one to encourage you.”* (Member from focus group 6)

This was discussed in the context of continuing to engage in the treatment of diabetes and the support received for depressive symptoms.

### 3.3. Structural Facilitators

Accessible Information

Sharing information was discussed as an important area of prevention and structural support. Participants mentioned how community health education seeking to facilitate treatment engagement must treat the individuals experiencing the condition as experts. One participant stated,

*“I think possibly if you, as educators, would pass on your knowledge to maybe somebody from the community who actually has gone through it, that way they can share their stories, share their progress. I think when people know other people who are close to them or take some role in their daily lives, I think it would encourage them to want to seek help or to hear a personal story from somebody that they know and how they changed their life. I think that would be helpful.”* (Member from focus group 8)

The sharing of information was discussed as a way to obtain help, not feel alone, and reduce fear related to help-seeking behaviors:

*“For people who might know something, it is super important to share information and keep sharing so they do not have these fears. Because there is a lot of fear in our community, we know this, with people with no immigration status, and this type of government that we have is so hard, right?”* (Member from focus group 1)

Accessible information was also mentioned for how the community could be a potential facilitator of access to care and adherence to treatment. Participants mentioned that support groups for diabetes and sharing stories with other people with diabetes or depression were helpful. Regarding depression, one participant reported,

*“You should see if there are support groups, churches, or people in the clinics. Ask to see if there are support groups because sometimes they have support groups that can help you. You hear other stories; you hear someone else’s, and sometimes it helps you because you see that you are not alone.”* (Member from focus group 3)

### 3.4. Personal Facilitators

#### 3.4.1. Family Support

Family support was discussed as a potential facilitator of sustained engagement in treating diabetes. Among the participants diagnosed with diabetes, the support of their families involved direct support, such as checking their blood sugar levels and reminding them to take their medicine. Additional support was provided, such as encouraging them to eat healthy foods, exercise, and be aware of their health status. The following quote illustrates how this support was offered to one of the participants:

*“My son has taught me a lot. He has helped me in that aspect... He says, ’No, Mom, it’s that we have to take care of ourselves. You have to take care of yourself.’ And the truth is, I have learned a lot from him because he always wants healthy things. ...he always does so much exercise. He is always busy exercising. He says to me, ’Mom, you can do it,’ he says, ’We are going to eat salads.’ With his plates of salmon and salad...”* (Member from focus group 1).

Participants who knew someone diagnosed with diabetes described providing support as being aware of the health of the other person and learning a new way of life, such as eating the same diet as the other person. One person described his experience with his significant other:

*“I used to be ignorant about hypertension or diabetes until I married B. I learned what is part of having discipline and a new way of life because she weighs less, and together, we are learning. I had a sweet tooth. To support her, I had to convert to half-and-half, some vegetarian, and a little carnitas, and I was removing it little by little. ...now I know that when her skin is dry, which is one of her symptoms, I can see her eyes, and I know the first thing I ask her is, ’Did you check?’ ’How was it today?’ When it [sugar level], is low, we can enjoy ourselves. When it is high, I have to skip a snack too. Because if one of you is going through the disease and the other doesn’t have it, we have to for the team. It is a change in our social life.”* (Member from focus group 2)

When comorbidity with depression was discussed, the participants mentioned that their families supported them by distracting them from being depressed. One participant said,

*“I have gone through many depressions. I am not going to say that they went away; from time to time, I fall into depression, but a mild one. He [my spouse] said to me, ’We are going to this place’. We went out, and it distracted me. I forgot. Depression is something tremendous because I even thought about hanging myself in the tree on my patio so that they would find me there. I thank God because he took away the bad things I was thinking about doing.”* (Member from focus group 2)

The participants mentioned both direct support and support that gave them intrinsic motivation. The motivation to keep going and get better came from one’s desire to continue to live a good life for their family. A participant who suffered from depression shared that her son was her reason to keep moving forward.

*“When I got pregnant with my son, I took my son to the psychologist, and I said, ’Look, I have this little angel that God sent to me to continue forward, and this will be my motor, this will be my psychologist, this will be my help.’ It was like I had come out ahead with his child, and sometimes I may trip and fall, but I always get up. But I have never taken the decision to commit suicide, no. Suicide, never.”* (Member from focus group 1)

Due to financial barriers associated with the cost of therapy (e.g., USD 200 per session) and fears of being considered a public charge, as she went through the immigration process, this person did not continue seeing the psychologist. A public charge is “an alien who receives one or more public benefits for more than 12 months, in total, within any 36-month period” [24].

#### 3.4.2. Spirituality

Different focus groups brought up conversations about faith in God that helped participants through difficult times. The participants mentioned how their faith served as both inspiration and support. One participant said,

*“I believe in my faith and the strength that God gives me because I get up early and don’t feel as tired...”* (Member from focus group 7)

In addition, the participants described their belief in God as helping them with their illnesses. One participant discussed their faith in God, even when barriers were present.

*“As I am now, if I get sick, the best doctor I have is God. I give him all my problems and all my illnesses. I want to make myself drink my juices... I do go to the doctor, but my barrier is that I cannot go alone because I don’t know how to get in a car and leave. It takes my daughters to get me to move. That is difficult for me, but I always go to church and ask God, ’Lord, I give myself to you because you are my best company. What I am taking is for my own good, for me to feel better’. Give everything to God so that God can help us.”* (Member from focus group 6)

Some participants referred to God as a doctor:

*“We have the opportunity to go to the best known and most effective doctor in the world, who I think is God.”* (Member from focus group 4)

However, others still mentioned their faith in God for opportunities such as work, treatment, and obtaining a good doctor. One participant said,

*“...thank God today I am standing because I asked God to give me a good doctor. God gave me and thank God and the doctor because science is there because God has left it for us. Thank God, and to her, I am on my feet...”* (Member from focus group 3)

## 4. Discussion

This study aimed to explore the experiences of Latino community members living in an urban setting regarding diabetes and co-occurring depression and the factors that shape their access to care and treatment engagement. These findings add to the growing body of research on healthcare access and utilization among Latinx in the U.S. [14,16,25,26,27,28,29], particularly with our focus on a familiar but less attended context of comorbid conditions of diabetes and depression. Results from the qualitative focus groups revealed that Latino participants faced significant challenges, such as financial burdens, a lack of health insurance, limited access to information, difficulty navigating healthcare institutions, fear, personal responsibility, and family dynamics. In addition, accessible information, community support, family support, and spirituality were identified as facilitators in improving healthcare access and treatment engagement. Previous qualitative studies used the social cognitive theory [30] or the social-ecological model as a framework by focusing on the social determinants of health [31]. When viewed through the social determinants of health framework [32], these findings emphasize the complex interplay of social, economic, and environmental factors that contribute to the structural barriers Hispanic participants face in accessing and engaging in healthcare. One qualitative study examined the barriers to and facilitators of help-seeking behavior regarding depression care among Latinxs with comorbid diabetes and major depression [33]. Like our findings, participants reported barriers such as language, stigma, and limited information. However, there were a couple of differences in the barriers, such as, in our study, the fear of receiving multiple diagnoses and not being able to maintain treatment. In the study by Hansen and Cabassa (2012) [33], the participants mentioned some fear of antidepressants as they were viewed as being potentially addictive and harmful. Additionally, there was limited mention of structural barriers such as financial and immigration status. Our findings were able to expand on the previous literature by providing a discussion about the structural barriers and other personal barriers, such as fear, that were not captured in the study focusing on help-seeking behavior regarding depression care among Latinxs with comorbid diabetes and depression.

### 4.1. Barriers to Care

#### Structural Factors

Our findings suggest that Latinos residing in urban communities continue to face structural barriers to healthcare, such as financial burdens, complex healthcare institutions, and a lack of access to information. The experiences of participants regarding the financial burden of healthcare as a barrier to care indicate that despite the Affordable Care Act expanding Medicaid eligibility, many Latinos continue to lack health insurance and face high healthcare costs. According to the U.S. Census Bureau’s report, Latinos continue to have the highest uninsured rates of any race or ethnicity in the U.S. [34]. Additionally, Latinos are more likely to live in states that have not expanded Medicaid eligibility as part of the Affordable Care Act, which further limits their access to healthcare services [35].

The experiences of uninsured participants underscore how the overlap of a lack of healthcare coverage and the high cost of medical care in the U.S. impact their decisions on whether to seek treatment and continue to engage with healthcare providers. Those who are uninsured and have no means to pay high out-of-pocket costs often delay seeking medical care, leading to more severe health problems in the long term. Extant studies have shown how uninsured individuals may be less likely to receive preventive care and may be more likely to delay seeking care until their condition becomes more severe [36].

Participants highlighted the nuances of health-related decision-making when individuals must choose between seeking preventive medical care or “putting bread on the table for the family.” They underscored how economic hardship can contribute to racial and ethnic disparities in healthcare access. Moreover, the participants discussed how the high cost of healthy food options and their limited income hinder their motivation to adopt healthy eating habits to prevent and reduce health risks. This challenge is particularly pronounced for Latinos living in urban areas that are also food deserts lacking accessible and affordable sources of fruits, vegetables, whole grains, low-fat milk, and other foods that comprise the full range of a healthy diet [37].

Prior studies have suggested that physical location, distance, and transportation can be significant barriers to accessing healthcare services [17]. However, contrary to these findings, our research did not reveal these factors as barriers to accessing healthcare services. This may be due to the high number of ambulatory health care providers, community health centers, and hospitals dispersed widely in the metropolitan area where the study was conducted. Our study found that type of employment is a more significant barrier than geography or means of travel. Latinos in the U.S. comprise a large share of the low-skilled labor force and are disproportionately employed in low-wage jobs [38] that offer little flexibility for taking time off during the day. Healthcare availability during traditional daytime hours of operation often limits access and treatment, and the opportunity for extended hours may improve service utilization [39].

Our findings demonstrate persistent knowledge gaps and needs among Hispanic families. Access to information is a significant challenge for Hispanic participants, particularly in terms of understanding diabetes and its treatment options. Uninsured and undocumented participants experiencing economic hardship often feel fearful of reaching out to medical providers and feel trapped between the need for treatment and the fear of deportation.

Hispanic patients with diabetes are also less likely to have access to diabetes education programs and self-management support [40], which can contribute to poor glycemic control and an increased risk of complications. Although health centers make brochures available in Spanish, the information is often insufficient, as observed among study participants who expressed uncertainty about their diabetes diagnoses and unanswered questions about treatment options and risks. These gaps highlight the shortcomings of the classic public health model of top-down information dissemination despite efforts to make materials more accessible and targeted. While informal support systems continue to fill these knowledge and support gaps, the results of this study suggest the value of including the perspectives of Latino patients with chronic health conditions in developing health literacy materials. Incorporating the voices of Hispanic patients and their support systems could help achieve some of the health impacts sought by health clinics and agencies using targeted dissemination as a public health strategy. While it is essential for policymakers and communities to increase mental health literacy and knowledge of treatment options, partnering with informal care may bolster a willingness within Latino communities to seek and engage in healthcare [25].

Consistent with the literature, participants’ experiences in the U.S. healthcare system highlight the difficulty of navigating a complex and fragmented system that discourages patients from seeking care and treatment [41]. Participant reports of discrimination while navigating health institutions add another layer of difficulty to the confusion they experience while attempting to seek care. Patients may also face unexpected costs and administrative obstacles when accessing care. Furthermore, research suggests that healthcare providers may approach uninsured patients differently than insured patients [42], influencing treatment decisions. For example, uninsured patients may be less likely to receive recommended treatments or more likely to receive lower-quality care than insured patients [43].

### 4.2. Personal Factors

Fear can be a significant barrier to seeking treatment for diabetes and comorbid mental health challenges [44]. Participants underscored how fear shapes their help-seeking behaviors, especially in the early stages of the disease. While part of the fear is related to migrant status, a common theme was the fear and misconceptions of being diagnosed with comorbid conditions. The fear of being diagnosed with diabetes and comorbid conditions of heart disease and mental illness is further worsened for those who do not have the resources to seek further treatment. This behavior is consistent with findings from past studies showing how fear and misconceptions can lead to the avoidance of treatment and poor diabetes control [45].

In addition, personal responsibility was a common theme across the focus groups. There was a recognition that, as Latinos, they were not socialized to seek treatment, especially as a preventive approach or during the early stages of disease. Research has identified the idea that one must persevere through personal hardships without help or that adversity builds character [25]. Even though preventive healthcare plays a crucial role in detecting potential health issues early and reducing the risk of chronic diseases, Latinos in the U.S. are less likely to engage in preventive healthcare compared to other racial and ethnic groups. Cultural norms among Latinos around health care access and treatment encourage the underutilization of preventive services and the seeking of care only when a disease requires urgent medical attention. Many of the study participants engaged in self-blame, and there was a lack of recognition of how multiple factors, such as economic hardship, discrimination, and a lack of health insurance, could exacerbate the lack of motivation to seek help and engage in treatment. Only one focus group discussed how what appears on the surface as a lack of interest in seeking treatment is a byproduct of the other barriers they face, such as poverty, being uninsured, and having to face more pressing concerns as immigrants or undocumented migrants. This finding underscores the need for culturally tailored education and community-based self-management programs that consider the socio-cultural roots of the lack of interest in preventive care and continued engagement with treatment.

Consistent with findings from previous studies regarding help-seeking stigma among Latinos, our results implicate stigma-related family dynamics as a significant barrier to healthcare access and treatment engagement. Participant reports of criticism from family members when seeking professional help allude to the strong influence of *familismo*, or the strong emphasis on family interconnectedness and loyalty, as a core cultural value [46]. Families that hold stigmatizing beliefs about specific health conditions, mental health issues, or healthcare services may be less likely to access care and engage in treatments [47]. These attitudes can contribute to a reluctance to seek professional help or disclose health concerns within the family. For the study participants, the opinions of family members perpetuate the ongoing stigma of seeking help, dissuading them from engaging in treatment. Foreign-born Latinos may even experience more significant stigma and concerns about the treatment of mental health disorders than U.S.-born Latinos, contributing to their reluctance to use specialty mental health services [48].

The lack of a robust support system within a family can negatively influence treatment engagement [25]. The participants made the connection between their capability to achieve better health and wellbeing and the amount of support they receive from their partners. It is important to note the gendered dimension of family support, where women-identified participants shared the emotional toll of lacking support from their partners. It was evident in the conversations that women continue to engage in their culture-bound caregiving roles to provide support to their men-identified partners. However, they also must contend with a lack of receptivity on the part of their partners and the support they need to cope with their health concerns.

### 4.3. Facilitators to Care

The accounts of participants in our study corroborate what prior studies have already found regarding how accessible information can facilitate treatment access and engagement [49]. Strengthening and promoting mental health awareness and education campaigns within local communities and organizations to increase healthcare literacy is needed to foster access and engagement [25].

Another important consideration is treating patients as experts in their own health experiences, which can provide valuable insights when developing community health education programs. Participants expressed that involving patients in developing community health education programs can help increase patient engagement and empowerment. They stressed the need for community educators with similar diagnoses and experiences.

Moreover, participant experiences highlight peer support as particularly beneficial for individuals with depression. Community organizations, such as churches, were reported as points of contact for peer support, which are often missed in intervention planning to address high rates of diabetes in Latino communities. Participants placed a strong emphasis on the role of peer support in accessing information and coping with a diagnosis, suggesting that peer supporters can help improve the scalable delivery of interventions, be acceptable and cost-effective adjunctive sources of support in complex interventions, and increase service outreach to populations with poor treatment engagement. The agreement between the participants about the role of churches as points of contact presents an opportunity to develop partnerships between public health institutions and religious institutions and address community needs.

The importance of family cannot be understated. Research has found that Latino women are more likely to seek out healthcare care than men [25]. Notwithstanding that many Latino families adhere to a more traditional, patriarchal family structure, women exert an influence on healthcare decision-making [25]. While elements of family dynamics have been identified as barriers to accessing and engaging in healthcare, families are complex systems, and the support of families and the role of women may offer opportunities to promote engagement through active partnering with families. Further, families may serve as partners in connecting more traditionally acceptable informal healthcare options with formal healthcare treatment. Few studies have included the experiences of the family members of individuals living with diabetes [40]. The perspective of family members is important to include in understanding the barriers and facilitators to help-seeking behavior and treatment adherence. Family dynamics has been previously recognized as both a support and a barrier; including family members’ perspectives provided deeper insights into their roles as facilitators or impediments. One participant mentioned not having information about hypertension but learned about it for their partner. Family members and other close friends might have limited knowledge of one condition but also of comorbid conditions and the intersecting relationship between having two conditions. Prevention programs and interventions could include teaching family members about conditions and treatment management for those who want to support their families. This could be a way that family members can shift away from being a barrier to being a facilitator.

### 4.4. Limitations

There are limitations to the current research study. First, only one of the eight focus groups was conducted in English, failing to encompass sample variability. It may have helped to have more representation from English-speaking Latinos. Including more English-speaking groups may offer more data on the barriers and facilitators experienced by more acculturated Latinos (e.g., English speakers), which may be beneficial to understanding their unique experiences based on language differences. Second, the context of this study was specific to a large southern urban area, and contextual circumstances may differ in other states, especially when we consider healthcare policies that vary from state to state. Third, due to time constraints in facilitating larger focus group discussions, not all participants had the opportunity to delve deeply into their experiences. The groups tended to identify barriers more prominently. Multiple sessions with each group could have provided more opportunities to explore the participants’ experiences related to the factors that improve their access to care and treatment engagement. Despite these limitations, this study offers an important contribution to existing gaps in understanding healthcare access and treatment engagement among Latinos experiencing comorbid conditions.

## 5. Conclusions

The current research study identified the structural and individual barriers and facilitators that Latino residents of Houston experience in relation to comorbid diabetes and depression. The core sub-themes for structural barriers included financial burdens, navigating healthcare institutions, and lacking access to information. The sub-themes under personal barriers included fears, personal responsibility, and negative family dynamics. The sub-theme under structural facilitators only included having accessible information, while the sub-themes under personal facilitators included family support and spirituality. Combining these experiences, it was evident that more advocacy is needed to ensure that healthcare institutions and providers take structural components into consideration to ensure that Latino patients and their families have more access to healthcare and better treatment adherence. The personal barriers and facilitators also informed the possibilities of reimagining how healthcare access is delivered with cultural appropriateness in mind.

## Figures and Tables

**Table 1 ijerph-21-00148-t001:** Key categories, themes, and sub-themes identified.

Key Category	Theme	Sub-Theme
Barriers to healthcare and mental health access and sustained treatment engagement	Structural Barriers	Financial burdens (i.e., healthcare, food costs)
		Navigating healthcare institutions (i.e., discrimination and racism)
		Lacking access to information (i.e., resources in Spanish)
Facilitators with healthcare and mental health access and sustained treatment engagement	Personal Barriers	Fears (i.e., immigration and health status)
		Personal responsibility (i.e., unhealthy eating behaviors)
		Family dynamic (i.e., unsupportive)
	Structural Facilitator	Having accessible information in communities and churches
	Personal Facilitators	Family support; spirituality

## Data Availability

The data are not publicly available for privacy and ethical reasons.

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
