# Peer review of "“It’s Not That We Care Less”: Insights into Health Care Utilization for Comorbid Diabetes and Depression among Latinos"

_ijerph, 2024, doi:10.3390/ijerph21020148_

Round 1
Reviewer 1 Report
Comments and Suggestions for Authors
This study explores healthcare access and treatment in the context of comorbid conditions. It is a qualitative study. It explores social support systems related to comorbid conditions and identifies barriers and facilitators with eight focus groups. Participants had a diagnosis of diabetes or depression or were close to someone with one of these conditions.
It is unclear what this study is really contributing. It is well established that personal barriers (e.g., financial, navigating healthcare institutions, fears) and facilitators (e.g., accessible information and family support) influence access to care. If someone had comorbid diabetes or depression, are they more likely/less likely influenced by these barriers and facilitators? Without a comparison group of people without diabetes or depression, how can you answer this question? Also, responses might be different if you have diabetes versus depression. To lump them together doesn’t make sense.
Some of the information mentioned in the text does not match the information in the citations used. I only checked a couple references, but enough to be concerned that other references may also be incorrectly cited.
Abstract, line 21. Change to “diabetes or depression”. Or do you mean people with both diabetes and depression? If such, what proportion of people with diabetes have depression? As you screened for participants, did you get this information?
Abstract, line 25. It is not clear what “structural” is referring to.
Introduction, line 36. The information in the text does not match the citation. The citation says 11.8%, not 17%. And the citation does not mention “6.5 million of the 60 million”
Introduction, line 41. The citation says “1.3” and not “1.5” and the reference is “non-Hispanic Whites” and not “non-Hispanic counterparts”
Introduction, line 46. No reference linked to 16.9%. This is much higher than other references noting depression for Hispanics. (see https://www.cdc.gov/nchs/data/databriefs/db303.pdf)
Introduction, line 56. What is meant by “elevated vulnerability of Latinos”? Also, what does “its” refer to. This whole sentence does not make sense.
Introduction, line 58. This suggests the study will consider several types of mental illnesses, not just depression. Is that what you meant? There is no mention of mental illnesses like anxiety, ADHD, OCD, bipolar disorder, schizophrenia, etc. in the paper.
The sentence “Knowledge of mental health conditions co-occurring with diabetes could advance our understanding of the role of mental health in patient access to care, especially for Latinos at the intersection of health, economic, and social disparities.” sounds nice, but this paper did not consider “mental health conditions”.
Introduction, line 62. You have not correctly established the prevalence of depression among Latinos and what health and behaviors health comorbidities are you talking about here? To suggest depression is comparatively high in Latinos compared with other ethnic/racial groups is incorrect. For example, non-Hispanic Blacks have much higher rates of depression. (see https://www.cdc.gov/nchs/data/databriefs/db303.pdf)
Introduction, line 74. But they are lower than for American Indians. Why didn’t you mention that?
Introduction, line 95. It is unclear what you are after in saying “perceive the comorbid conditions of diabetes and depression.” Perceive in what sense? Is it a bad or unfortunate thing to have diabetes and depression? Of course.
Introduction, lines 98-100. It should be emphasized that the results apply to Latinos with diabetes “and” depression. This has implications as to who you can generalize the results. Unfortunately, there is not a comparison group to give the results greater meaning.
Materials and Methods, line 103. Now you say the comorbid condition of interest is depression. Isn’t it both diabetes and depression”
Sample, line 113. Why eight focus groups? It reads “closely associated with someone” that has both diabetes and depression. So, you couldn’t participate if you were closely associated with just diabetes or just depression. How did you confirm that someone had diabetes? Depression?
Does the fact that there was only one English-speaking focus group mean that most of your participants only spoke Spanish? This will have an impact on access to healthcare. This fact may confound your results.
Conclusion, line 600. Do you mean comorbid diabetes and depression?
Comments on the Quality of English LanguageAdequate.
Reviewer 2 Report
Comments and Suggestions for Authors
This qualitative study addresses the obstacles and barriers faced by Latinos in accessing healthcare in conditions of comorbidity, namely in people suffering from both diabetes and depression.
I would like to offer some suggestions that I think might enhance the clarity and methodological strength of the work.
- 1. The authors state that they included in the study both patients with comorbidity and patients who have a close person suffering from depression. This sets up two quite different experimental conditions, and it would be important to explain how many people fall into each category and justify the rationale for this choice. For example, explaining that even those who are not subjectively involved in depressive symptoms can still contribute to explaining the difficulty in accessing care.
- 2. There are no notes in the method that allow understanding if qualitative type software, for example, Atlas ti or MAXQDA, were used. If the data were entirely manually coded, this must be explained in the limitations by referring to the biases that may have contributed to the attribution of codes and the identification of themes.
- 3. What really lacks in the discussion of the results is the comparison with other similar studies. The paper is well-argued and addresses an important topic, but what can we say compared to other populations of diabetic patients? In my experience, I can say that even where the health system is completely free (Italy), many patients refuse care. I believe that this has to do with health beliefs and “Health Beliefs”. It would be important to use this theoretical model to explain these behaviors.
- 4. As for access to care in relation to depression, it is important to also involve the concept of Stigma towards mental health.
- 5. It would be important to conclude the paper also with some ideas on how the authors would intend to solve this problem and what policies could be implemented to improve access to both physical and psychological care for this community.
- I hope to have provided useful feedback, good luck with your work.
No issue detected
Round 2
Reviewer 1 Report
Comments and Suggestions for Authors
The authors responded to my comments/concerns. A final request is that the title be fixed. The first part "It's not that we care less" could be deleted, as it is a bit awkward and unclear.
Comments on the Quality of English LanguageThe overall paper could use an editor.